# Improving the Detection of the Contact Point in Active Sensing Antennae by Processing Combined Static and Dynamic Information

**DOI:** 10.3390/s21051808

**Published:** 2021-03-05

**Authors:** Luis Mérida-Calvo, Daniel Feliu-Talegón, Vicente Feliu-Batlle

**Affiliations:** 1Instituto de Investigaciones Energéticas y Aplicaciones Industriales, Universidad de Castilla-La Mancha, 13071 Ciudad Real, Spain; Luis.Merida@alu.uclm.es; 2Robotics, Vision and Control Group, Universidad de Sevilla, 41092 Sevilla, Spain; Danielfeliu@us.es; 3Escuela Técnica Superior de Ingeniería Industrial de Ciudad Real, Universidad de Castilla-La Mancha, 13071 Ciudad Real, Spain

**Keywords:** robotic sensor, obstacle recognition, sensing antenna, flexible robot, motion control, active vibration damping, impact detection

## Abstract

The design and application of sensing antenna devices that mimic insect antennae or mammal whiskers is an active field of research. However, these devices still require new developments if they are to become efficient and reliable components of robotic systems. We, therefore, develop and build a prototype composed of a flexible beam, two servomotors that drive the beam and a load cell sensor that measures the forces and torques at the base of the flexible beam. This work reports new results in the area of the signal processing of these devices. These results will make it possible to estimate the point at which the flexible antenna comes into contact with an object (or obstacle) more accurately than has occurred with previous algorithms. Previous research reported that the estimation of the fundamental natural frequency of vibration of the antenna using dynamic information is not sufficient as regards determining the contact point and that the estimation of the contact point using static information provided by the forces and torques measured by the load cell sensor is not very accurate. We consequently propose an algorithm based on the fusion of the information provided by the two aforementioned strategies that enhances the separate benefits of each one. We demonstrate that the adequate combination of these two pieces of information yields an accurate estimation of the contacted point of the antenna link. This will enhance the precision of the estimation of points on the surface of the object that is being recognized by the antenna. Thorough experimentation is carried out in order to show the features of the proposed algorithm and establish its range of application.

## 1. Introduction

The first works on tactile sensors can be traced back to the early 1990s, e.g., [1]. However, natural tactile sensors, such as whiskers and antennae, have been explored in more recent years, e.g., [2]. In the last fifteen years, several attempts have been made to build biomimetic active sensory applications, which are also known as vibrational systems. Mammal- or insect-inspired sensing has led to the appearance of several engineering applications, such as the work of [3], in which whisker-based texture discrimination on a mobile robot was presented.

In the last two decades, a robust and compact sensor device has been developed that mimics the antennae that many insects have. This is called a “sensing antenna” and is an active sensor that consists of a flexible link moved by servo-controlled motors and a load cell placed between the beam and the motors. This device replicates the touch sensors that many insects have and carries out an active sensing strategy in which the servomotor system moves the beam back and forth until it hits an object. At this instant, information regarding the motor angles combined with the force and torque measurements makes it possible to calculate the positions of the hit points, which is valuable information about the surface of an object. A 3D map of the surface of that object can then be obtained, which allows its recognition. The design of efficient artificial sensing antennae has been addressed by several researchers, as in [4], in which issues such as the minimum number of sensors required and the precision attained using antennae with different profiles (cross- sections and curvatures) were studied.

Lightweight whiskers are being developed with several geometries and are being used in different applications like object identification and spatial localization [5], Simultaneous Localization And Mapping techniques (SLAM) for robot navigation [6], classification of objects based on their material properties [7], underwater sensing whiskers to measure water flow velocity [8], whiskers for fluid velocity sensing [9], with the purpose of using them as an aid for drone navigation in dark or turbulent environments, and soft whiskers that actively adjust their morphology in order to regulate their sensitivity [10].

Since a sensing antenna includes a flexible link, its dynamic model is quite complex, because it is highly nonlinear and of infinite dimensions, i.e., it has an infinite number of vibration modes. Fortunately, the amplitude of the modes decreases as their vibration frequencies increase. The antenna dynamic model can, therefore, often be truncated, and only the two or three lowest vibration modes are maintained. Higher vibration modes can consequently be ignored and regarded as noise.

Several approaches with which to analyze the data obtained from a tactile sensor hitting an object have been proposed, e.g., [11,12]. Of the information that can be obtained from these data, the accurate detection of the impact instant and the point of the beam at which the antenna collides with the object are of the utmost importance when attempting to reduce the time required to estimate the 3D coordinates of the points on the object’s surface with which the antenna comes into contact and improve the estimation efficiency and accuracy.

Efficient algorithms with which to detect the collision of a robot with an object are already available. They are based on detecting when either a measured torque or its residue, i.e., the difference between the measured torque and its value predicted by a model, exceeds a threshold; see [13] for rigid and [14,15,16] for flexible link robots. Efficient algorithms based on the same principle have recently been developed that detect the instant at which a sensing antenna impacts an object. The algorithm proposed in [17] detected this instant with a delay of about 6 ms. It was later improved in [18], reducing this delay to about 4 ms. These algorithms are required in order to control robots that interact with the environment. For example, in [19], one of these algorithms was used to achieve force control with robustness to uncertainties in the environment.

The objective of this paper is to improve the accuracy of the algorithms that estimate the point of contact between the antenna and the object. These algorithms should work only in contact mode. They are then activated at the instant at which the contact is detected by one of the algorithms mentioned in the previous paragraph. Moreover, in order to improve the accuracy of this contact instant estimation, the control system in charge of moving the antenna in the free space must remove the vibration of the antenna during the maneuver [17].

The starting point for this research is defined as: (1) the servo controlled antenna developed in [20]; (2) the control system developed in [21] to remove vibrations in the free movement mode of the antenna; (3) one of the mechanisms described in [17] to estimate the impact instant of the antenna; and (4) some already existing algorithms that estimate the contact point, which will be presented in Section 2.

This paper is structured as follows. Algorithms that have already been used to estimate the contact point of a sensing antenna are presented in Section 2. Section 3 describes our sensing antenna prototype, while Section 4 shows the development of our new contact point estimation algorithm, which is the contribution of this paper. Section 5 reports several experiments, which illustrate that the estimations provided by our algorithm outperform previous results, and Section 6 discusses the results obtained and compares them with previous results. Finally, Section 7 presents some conclusions.

## 2. Related Work

This section describes the methods that already exist to determine the point at which the antenna comes into contact with an object. Two of these methods will be highlighted, since they are the basis of our approach.

The first research on this subject implemented a sensing system composed of a flexible link that moved on a horizontal plane, a torque sensor, a joint position sensor, a rotational actuator and a payload at the tip end of the link. In [22], this setup was used to determine the contact point of the antenna by measuring the lowest natural frequencies of the oscillations of the link during contact mode. A schematic representation of this methodology is shown in Figure 1a. It was proven that the contact position cannot be determined if only the fundamental natural frequency of the beam is taken into consideration. It was subsequently shown that the information regarding the fundamental and the second natural frequencies is sufficient to determine the contact point, provided that: (1) the link has a uniform mass and stiffness distributions; and (2) a light payload is added to the endpoint of the antenna in order to avoid indeterminacy in the estimation of some points of the link. The frequencies are then mapped onto a contact-point coordinate function. Experiments demonstrated that this is a very useful sensing strategy and reported estimations of the contact point of the link with relative errors of about 3% of the length of the antenna.

Another approach was shown in [11], in which use was made of a sensing antenna composed of a flexible link, two rotational motors that generated 3D movements, joint position sensors installed in the motors and a six-axis force/moment sensor placed at the base of the link. The basic idea is that the contact point of the antenna can be determined as the quotient between the measured moment and force. This paper made corrections to this estimator, taking into account gravity and lateral slip. Three active motions are needed in order to obtain contact point estimations with relative errors of between 1% and 5% of the length of the antenna.

A sensing whisker was implemented in [23]. Its setup was similar to that of [11], but a link that was much more flexible than those used in previous works was utilized, as shown in Figure 1b. This link underwent geometrically nonlinear deflections, which complicated the estimation of the contact point obtained from measurements of moments and forces at the base of the whisker. The nonlinear model of elastica was used, which was numerically reproduced in 3D. Experiments concerning contact imaging recognition were carried out with this whisker, which yielded a resolution of 15.1 mm using a flexible link of 455 mm. This implies relative estimation errors of the contact points of about 3% of the length of the antenna.

Another approach is that of employing a tapping strategy in sensing distance information by inspecting the strain according to a specific angular displacement [24]. The authors of the paper in question developed a soft active whisker sensor that yielded errors of over 5%.

An antenna prototype similar to [11] was developed in [20], but it had a payload at the tip. This prototype combined an estimator of the impact instant with an estimator of the contact point based on the static information provided by the forces and torques measured by a load cell. Contact point estimation experiments were carried out in [25]. The resolution obtained was 14 mm for a 505 mm flexible link, i.e., a relative error of 3%. In [17], the tip payload was subsequently removed, and its link was replaced with a link that was almost twice as long. An estimator based on the quotient between the torque and the force measured at the base of the antenna was used, which yielded a resolution of 30 mm using a link of 980 mm in length, i.e., a relative error of 3% of the length of the antenna.

Finally, we should mention a recently conceived of algorithm that uses a nonlinear dynamic estimator to determine the contact position in a massless link with a payload at the tip [26]. This algorithm is simple, but has the drawbacks that its convergence (stability) has not been proven and that only simulated results have been provided.

In this paper, therefore, we pursue the creation of a contact point estimator with the following features:It does not require a small payload placed at the tip of the antenna. This will allow the antenna to move faster and obtain more information about the object surface in a given period of time, since the load to be moved by the actuators will be reduced.The accuracy of the estimation must be better than the accuracy provided by the aforementioned methods, i.e., a relative error that is much lower than 3% must be attained.The algorithm must be sufficiently fast and reliable to be used in a real prototype.

In order to obtain these features, the following sections show the development of an algorithm that is based on the adequate fusion of the information provided by two of the previous strategies: that which uses the natural frequencies of vibration of the antenna [22] and that which uses the static information provided by the forces and torques measured at the base of the antenna [17]. This methodology is schematically represented in Figure 1c.

## 3. Materials

### 3.1. Experimental Setup

The experimental prototype is a two-degrees-of-freedom (2DOF) robotic system with a single flexible link, which is used as a sensing antenna in haptics applications, shown in Figure 2a. Its structure is made of stainless steel, with three legs to ensure perfect stability. The flexible link is attached at one of its ends (denoted as the absolute origin *O*) to two direct current (DC) mini-servo actuator PMA-5A motor sets that include harmonic drive reduction gears. One servo-motor rotates the system with azimuthal movements, whereas the other rotates the system with elevation movements. These DC motors have incremental optical encoders that measure the angular position of the motors $\theta_{1}$ and $\theta_{2}$, which are the azimuthal and attitude joints, respectively. The system also has a force-torque (F-T) sensor located at the origin *O*, which allows the measurement of the Cartesian coupling force and torque existing between the link and the servomotor structure. The signals are acquired through the use of gauges located inside the sensor, which are multiplexed and amplified in order to send the information regarding forces and torques to the data acquisition card (DAQ). The servomotor structure, which rotates according to the motor angles, holds the F-T sensor and the flexible beam that is attached to one of its two sides. More information about this platform can be found in [27].

A schematic diagram of the system is represented in Figure 2b, in which the equivalent length of the beam is *l*; $P_{t}$ is the tip of the flexible beam, and $P_{r}$ is the beam tip itself when the beam is considered to be a rigid beam; ΔP is a 3D vector that describes the beam deflection at the tip; *E* is the Young modulus; *I* is the inertial moment resulting from the flexible beam cross-section; and $m_{t}$ is the tip mass, which will be removed in our experiments. The tip position is expressed in spherical coordinates $\phi_{1}$ and $\phi_{2}$ with regard to the absolute frame *O*-$X^{a}Y^{a}Z^{a}$, along with the motor angles $\theta_{1}$ and $\theta_{2}$, which also define the orientation of the *O*-$X^{m}Y^{m}Z^{m}$ frame. Coupling forces and torques are provided by the F-T sensor in this frame and are represented by ${\vec{F}}^{m}=\left(F_{x}^{m},F_{y}^{m},F_{z}^{m}\right)$ and ${\vec{\Gamma }}^{m}=\left(\Gamma_{x}^{m},\Gamma_{y}^{m},\Gamma_{z}^{m}\right)$, respectively.

A third frame, $X^{c}Y^{c}Z^{c}$, is defined from the force measured by the F-T sensor at the point at which the antenna touches the surface of an obstacle at a certain point of the antenna at a determined instant: (1) the $Y^{c}$ axis is the direction of the reaction force of the object on the antenna; (2) the $Z^{c}$ axis is perpendicular to $Y^{c}$ and to the vector $\vec{OP_{r}}$ (see Figure 2b); and (3) the $X^{c}$ axis is perpendicular to the other two axes. Figure 3 shows all the frames and the relations among them.

The data acquisition and control algorithms were programmed using Labview. Simulations were performed using Simulink/ MATLAB. The data acquisition (measurements, control signals and written data) sampling time was $T_{s}=1$ ms. A scheme of the components of the system is shown in Figure 4.

Table 1 shows the parameters of the two motors of the system, where the subscript i=1,2 defines the two degrees of freedom of the antenna. $K_{mi}$ is the motor constant that defines the motor torque applied on the basis of the input voltage; $J_{i}$ is the motor rotational inertia; $v_{i}$ is the viscous friction coefficient; $V_{i}^{nlc}$ is the Coulomb friction in terms of voltage; and $n_{i}$ is the reduction gear ratio.

Links of different diameters and the lengths and loads at their tips can be used on this platform. However, there are some constraints, such as the size of the antenna and the load at the tip, owing to mechanical limitations: the torques of the motors, the maximum speed and acceleration of the movements and the yield stress of the material of which the antenna is composed. In this work, a very long and lightweight flexible link made of carbon fiber and without any load at the tip ($m_{t}=0$) was used for two reasons: (1) since the workspace of the antenna is directly related to the length of the antenna, we chose a long antenna in order to reach more distant points; and (2) adding a mass to the tip is mechanically unnecessary and slows down the motion of the antenna, while the use of a lightweight antenna without any load at the tip makes it possible to perform faster movements owing to the reduced overall mass of the antenna. Table 2 shows the characteristics of the sensing antenna, where *E* is the Young modulus, *r* is the radius of the link cross-section and *I* is the inertial moment of the link cross-section, which can be calculated from *r* by taking into account the fact that the link has a circular section.

A photograph of the robotic sensing antenna is provided in Figure 5. It shows a camera-based system, which is used as an external sensor to measure the position of the tip of the flexible link.

The dynamic models of these subsystems are defined below. Dots are used to denote differentiation with respect to time *t*, and primes denote partial differentiation with respect to coordinate *x*.

### 3.2. Actuator Dynamics

Our 2DOF sensing antenna is moved by two DC motors with high ratio reduction gears. The dynamic model of each motor *i* (i=1,2 is the azimuthal or attitude joint) with a reduction gear of ratio $1:n_{i}$ is provided by:(1)$$\Gamma_{i}=n_{i}\cdot K_{mi}\cdot u_{i}=J_{i}\cdot {\ddot{\theta }}_{i}+\nu_{i}\cdot {\dot{\theta }}_{i}+\Gamma_{i}^{nlc}+\Gamma_{i}^{coup}$$
where $\Gamma_{i}$ is the motor torque, $u_{i}$ is the input voltage to the motor, ${\ddot{\theta }}_{i}$ and ${\dot{\theta }}_{i}$ are the acceleration and velocity of each motor (where $\theta_{i}$ is the angle of the motor *i*), $\Gamma_{i}^{coup}$ is the moment transmitted by the actuator to the link and $\Gamma_{i}^{nlc}$ is the Coulomb friction term, which is non-linear.

In this equation, voltages $u_{i}$ are the control signals. As it is assumed that the motors have servo amplifiers with very fast dynamics, the currents of the motors and, therefore, the motor torques $\Gamma_{i}$ are assumed to be proportional to the previous voltages. The constants $K_{mi}$ define that proportionality. Table 1 shows the values of the parameters of Equation (Equation 1).

### 3.3. Planar Dynamics of the Contact Sensing

Figure 6 shows a scheme of the antenna in contact mode. It represents the bending of the antenna in a plane perpendicular to the surface of the object at the contact point (this plane includes the origin *O* and the contact point). In this figure, *y* is the deflection of the antenna, *t* is time, XY is the 2D frame of the bending plane, where *X* is the direction of the antenna if it is rigid, *x* is the position along the beam, $x_{c}\in \left[0,l\right]$ is the point of the antenna at which the contact is produced and $y_{c}$ is the deflection at that point, which is assumed to be constant because the contacted object is rigid.

The following assumptions are made about the antenna:The beam deflection is limited to 10% of the total beam length in order to obtain a linear beam deflection. In this case, deflections *y* are small when compared to the corresponding *x* values. The deflected abscissa, therefore, has approximately the same value as the undeflected abscissa, $arctan\left(\frac{y}{x}\right)\approx \frac{y}{x}$, and both *x* and *y* become the coordinates of a point of the deflected beam expressed in the frame XY.This assumption is justified because: (a) the figure of 10%, needed to assume a geometrically linear deflection, is an approximate value commonly used in studies of beam deflections and vibrations (a justification of this value can be found in [28]); (b) though free movements are performed carrying out fast trajectories, deflections are lower than this limit because the antenna is very lightweight, and then, the inertial forces are low; and (c) in contact mode, the force exerted by the antenna on the object is programmed to be high enough to allow a reliable estimation of the contact point, but low enough to prevent exceeding this deflection limit.The antenna was manufactured to have a uniform cross-section.Since the antenna is a very slender beam, rotatory inertia, shear deformation and internal friction are ignored.The total mass of the antenna is much smaller than that of the contacted object, and the object does not, therefore, move during the sensing process.Regarding the antenna dynamics in the free movement, it was demonstrated in [27] by carrying out extensive simulations and experimentation that, since the linear density of the beam ρ is sufficiently small: (a) the vibration associated with the first mode is much more relevant than the vibrations associated with the other modes; (b) the Coriolis and centripetal torques are much smaller than the inertial torques, and they can, therefore, be neglected; (c) the previous item allows us to assume that the azimuthal and attitude dynamics are approximately decoupled; and (d) the gravitational torque is significant in the attitude component of the movements.

The following assumptions are made about the contact mode:The contacted object is rigid.The contact is produced at a single point, as was commonly assumed in the previous works.Sliding of the contacting bodies relative to one another is negligible once a specific value of the pushing force has been attained, i.e., if the pushing force of the antenna on the object surpasses a specific value, slipping is prevented.Since there is no deformation in the *X* axis of the antenna, longitudinal contact forces do not influence the dynamics modeled.The linear density of the beam ρ is sufficiently small, and the contact force is sufficiently large to be able to assume that gravity influences neither the vibration modes of the beam nor its steady-state deflection. In particular, Section 5.6.2 checks that the first vibration frequency, which is the one used in this work, does not change because of the effect of gravity.

Linear dynamics are assumed for the system shown in Figure 6. The superposition principle can, therefore, be applied, and the deflection of the antenna can be expressed as the addition of two components:(2)$$y\left(x,t\right)=y_{t}\left(x,t\right)+y_{s}\left(x\right)$$

The first component $y_{t}\left(x,t\right)$ is the transient response of all the points of the beam, while $y_{s}\left(x\right)$ is the steady-state component (permanent deflection of the beam assuming that the transient has vanished).

If the third assumption about the antenna holds, the transient can be obtained from the partial differential equation of Euler–Bernouilli beams:(3)$$E\cdot I\cdot y_{t}^{⁗}\left(x,t\right)+\rho \cdot {\ddot{y}}_{t}\left(x,t\right)=0$$
from which the effect of gravity is removed.

Equation (Equation 2) holds because a linear dynamic system is assumed. Since the superposition principle is applied, the transient can be obtained from Equation (Equation 3) by assuming that the base end of the link is clamped to the motors and the point at which the link comes into contact with the object is aligned with the direction of the base end of the link, i.e., with $X^{m}$. The following contour conditions must, therefore, be taken into account in order to solve Equation (Equation 3) [22]:(4)$$\begin{array}{c} y_{t}\left(0,t\right)=0,\;y_{t}^{\prime }\left(0,t\right)=0,\;y_{t}^{\prime \prime }\left(l,t\right)=0,\;y_{t}^{\prime \prime \prime }\left(l,t\right)=R_{m}\cdot {\ddot{y}}_{t}\left(l,t\right),\\ y_{t}\left(x_{c},t\right)=0,\;y_{t}^{\prime }\left(x_{c}^{+},t\right)=y_{t}^{\prime }\left(x_{c}^{-},t\right),\;y_{t}^{\prime \prime }\left(x_{c}^{+},t\right)=y_{t}^{\prime \prime }\left(x_{c}^{-},t\right)\; \end{array}$$
where $x_{c}^{-}$ signifies that *x* approaches $x_{c}$ from the left, $x_{c}^{+}$ signifies that *x* approaches $x_{c}$ from the right and $R_{m}=m_{t}/m_{b}$ is the ratio between the mass of the payload and the mass of the beam, $m_{b}=\rho l$. In our case, $R_{m}=0$, because the link has no payload at the tip, i.e., $m_{t}=0$. Equations (Equation 3) and (Equation 4) can be solved by using the separation of variables method. This process, which was described in [29], yields a general solution of the form:(5)$$y_{t}\left(x,t\right)=\sum_{i=1}^{\infty }A_{i}\cdot \varphi_{i}\left(x\right)\cdot \sin\left(\omega_{i}\cdot t+\alpha_{i}\right)$$
where $\varphi_{i}\left(x\right)$ is the natural mode, $\omega_{i}$ is the angular frequency, $A_{i}$ is the amplitude and $\alpha_{i}$ is the phase angle of the i-th vibration mode.

Upon introducing the variable $\beta_{i}^{2}=\omega_{i}\cdot \sqrt{\frac{\rho }{E\cdot I}}$, the frequencies of the vibration modes are provided by the solutions to the equation:(6)$$\begin{array}{c} \cosh\left(\beta_{i}\cdot \left(l-x_{c}\right)\right)\cdot \left[\sin\left(\beta_{i}\cdot x_{c}\right)\cosh\left(\beta_{i}\cdot x_{c}\right)-\sin\left(\beta_{i}\cdot \left(l-x_{c}\right)\right)\right]+\\ \cos\left(\beta_{i}\cdot \left(l-x_{c}\right)\right)\cdot \left[\sinh\left(\beta_{i}\cdot \left(l-x_{c}\right)\right)-\sinh\left(\beta_{i}\cdot x_{c}\right)\cdot \cos\left(\beta_{i}\cdot x_{c}\right)\right]+\\ \cos\left(\beta_{i}\cdot x_{c}\right)\left[\sin\left(\beta_{i}\cdot x_{c}\right)-\cos\left(\beta_{i}\cdot x_{c}\right)\cdot \sinh\left(\beta_{i}\cdot x_{c}\right)\right]=0\; \end{array}$$
which are obtained by taking into account that $R_{m}=0$. A detailed description of how this model was obtained can be found in [22].

The second component $y_{s}\left(x\right)$ can be obtained from the Euler–Bernouilli deflection equation of a static beam, ignoring the effects of gravity:(7)$$y_{s}^{⁗}\left(x\right)=0$$

The following contour conditions have to be taken into account in order to solve this differential equation:(8)$$\begin{array}{c} y_{s}\left(0\right)=0,\;y_{s}^{\prime }\left(0\right)=0,\;y_{s}^{\prime \prime }\left(l\right)=0,\;y_{s}^{\prime \prime \prime }\left(l\right)=0,\\ y_{s}\left(x_{c}\right)=y_{c},\;y_{s}^{\prime }\left(x_{c}^{+}\right)=y_{s}^{\prime }\left(x_{c}^{-}\right),\;y_{s}^{\prime \prime }\left(x_{c}^{+}\right)=y_{s}^{\prime \prime }\left(x_{c}^{-}\right)\; \end{array}$$

The solution of Equation (Equation 7) is a three degree polynomial in *x*. The contour conditions of Equation (Equation 8) impose different polynomials on the intervals before and after the contact point $x_{c}$:(9)$$\begin{array}{c} y_{s}\left(x\right)=-\frac{y_{c}}{2}\cdot {\left(\frac{x}{x_{c}}\right)}^{3}+\frac{3\cdot y_{c}}{2}\cdot {\left(\frac{x}{x_{c}}\right)}^{2},\;\left[0,x_{c}\right]\\ y_{s}\left(x\right)=\frac{3\cdot y_{c}}{2}\cdot \left(\frac{x}{x_{c}}\right)-\frac{y_{c}}{2},\;\left[x_{c},l\right] \end{array}$$

### 3.4. Steady-State Measurements

The variables measured by the F-T sensor can be obtained from the Euler–Bernouilli deflection equation of a beam at x=0:(10)$$|\vec{\Gamma }|=E\cdot I\cdot |y^{\prime \prime }\left(0,t\right)|$$
whose direction is normal to the plane defined by the frame XY and:(11)$$\left|\vec{F}|=E\cdot I\cdot |y^{\prime \prime \prime }\left(0,t\right)\right|$$
whose direction is opposite the pushing.

The fifth assumption of the antenna states that the internal damping of the beam is zero. The transient response, therefore, never vanishes. However, let us now assume that the transient $y_{t}\left(x,t\right)$ has disappeared after some time. This can be caused by the light internal damping of the beam, which always exists and which, as time goes by, makes the transient response vanish, or it may be the result of an active control that removes the vibrations. If we, therefore, assume that the transient $y_{t}\left(x,t\right)$ has vanished, only $y_{s}\left(x\right)$ remains, and the previous expressions become:(12)$$|\vec{\Gamma }|=E\cdot I\cdot \left|\frac{3\cdot y_{c}}{x_{c}^{2}}\right|$$
(13)$$|\vec{F}|=E\cdot I\cdot \left|\frac{3\cdot y_{c}}{x_{c}^{3}}\right|$$

## 4. Methods

This section presents the new method with which to detect the point of the antenna that comes into contact with an object. It is based on the fusion of the information provided by the transient component of the dynamic response of the system(Equation (Equation 5)) and the steady-state component of the response (Equation (Equation 9)). Characterizing these two components of the response from the measured signals makes it possible to take advantage of the separate benefits of these two kinds of information.

Our method estimates the contact point using the first natural frequency of the transient response, which is obtained from Equation (Equation 6). If there were two possible contact points for a given vibration frequency, the true point would be discriminated by using an estimator based on the steady state, which utilizes the measurements of Equations (Equation 12) and (Equation 13). It is important to note that the angular positions of the motors are controlled throughout the process.

The method proposed in order to estimate the contact point $x_{c}$ is synthesized in the following steps:The antenna moves freely in 3D space in a motor servo-controlled manner until it hits an object.An algorithm estimates the instant $t_{c}$ at which the antenna collides with the object.After contact has been made: (1) the motors of the sensing antenna keep moving in a servo-controlled manner in an attempt to follow the commanded trajectory; and (2) the reaction force of the object on the antenna, $-{\vec{F}}^{m}$, and the coupling torque, $-{\vec{\Gamma }}^{m}$, keep increasing as a consequence of the motor movement.The motors are stopped at the instant $t_{1}$ at which the antenna pushes the object with a programmed force whose magnitude is $F_{m}^{m}$. Hereafter, the base of the antenna will remain quiet, and according to contact mode Assumption 3, the contact point of the antenna and the object will remain steady without slipping. Possible rebounds of the antenna with the object as a consequence of the impact vanish before reaching instant $t_{1}$. Hereafter, a contact mode is reached, and the frame $X^{c},Y^{c},Z^{c}$, which is attached to the contact point, is defined from the force measured by the F-T sensor at this instant (see Figure 3). After $t_{1}$, the behavior of the antenna is characterized by the following:(a)Its vibrations are almost undamped, and their frequencies are provided by the solution of Equation (Equation 6).(b)There is no vibration in the $X^{c}$ direction, since Assumption 4 of contact mode implies that the antenna does not have compressive or tensile strain.(c)Antenna vibrations are consequently produced only in directions $Y^{c}$ and $Z^{c}$.(d)The pushing of the antenna often damps the vibration in the direction $Y^{c}$. In these cases, the only significant vibration is produced in the $Z^{c}$ direction.(e)The shape of the antenna in the steady state is provided by Equation (Equation 9) and is contained on the plane defined by the axes $X^{c}$ and $Y^{c}$.Once the motors have stopped, two estimators of the contact point are triggered at time $t_{1}$. The first estimator (denoted as Estimator 1) is based on calculating the first (fundamental) natural frequency of the remaining vibration and is active during a programmed time interval $\Delta t_{e1}$. The second estimator (denoted as Estimator 2) is based on calculating the torque/force ratio and ends when the torque/force ratio converges to a constant value. The time that has elapsed until this convergence has been produced is represented by $\Delta t_{e2}$.The estimation process ends at the instant $t_{e}=t_{1}+max\left(\Delta t_{e1},\Delta t_{e2}\right)$.At instant $t_{e}$, the procedure yields the position of the contact point. Of all the points provided by the estimator on the basis of the fundamental frequency (Estimator 1), the procedure proposes that which is closer to the point provided by the torque/force ratio estimator (Estimator 2).Finally, at instant $t_{e}$, the motor trajectories are reprogrammed in order to ensure that the antenna keeps pushing the object or starts searching for another point with which to make contact.

The algorithms involved in this method are described in the following subsections.

### 4.1. Motor Control

The antenna always moves in a controlled manner. The controllers do not change: their parameters remain the same in free and contact modes. Only the trajectories for the two motors $\theta_{i}^{*}\left(t\right)$, i=1,2 change once a contact point has been estimated.

The scheme proposed in [17,27], which is shown in Figure 7, is used to control the position of the motors. It includes a feedback of the coupling torque $\Gamma_{i}^{coup}$ (the torque measured by the F-T sensor at the base of the antenna) that makes the dynamics of the controlled motors insensitive to the movements of the antenna and to its operating mode (free or contact). This feedback of the coupling torque drastically simplifies the dynamic models used to design the motor controllers, thus making the design of the motor control system relatively simple.

Proportional, integral and derivative (PID) controllers with a low-pass filter term, i.e., of the form $R\left(s\right)=\frac{a_{2}\cdot +a_{1}\cdot s+a_{0}}{s\cdot \left(s+b\right)}$, are used in the motor control because they ensure good trajectory tracking, compensate disturbances such as unmodeled components of the friction and are robust to parameter uncertainties, thus providing a precise and fast positioning of the motor. These PID controllers are arranged according to the two degrees of freedom scheme shown in Figure 7, in which two of these controllers, $R_{1,i}\left(s\right)$ and $R_{2,i}\left(s\right)$, are implemented in each motor control in order to place the poles and zeros of the closed-loop system at the desired locations.

The four closed-loop poles of this scheme are placed in the same arbitrary location $p_{i}$ by following the algebraic method described in [27]. The two zeros of the closed-loop are also placed in $p_{i}$ in order to cancel two poles of the closed-loop. The overall transfer functions of the servo-controlled motor then become:(14)$$M_{i}\left(s\right)=\frac{\theta_{i}\left(s\right)}{\theta_{i}^{*}\left(s\right)}=\frac{1}{{\left(1+\epsilon_{i}\cdot s\right)}^{2}}\;;\;\;\epsilon_{i}=-p_{i}^{-1},\;i=1,2$$

Since very fast motor movements are desired, the absolute values of poles $p_{i}$ are chosen to be high.

### 4.2. Estimation of the Contact Instant

Estimating the instant at which the contact is produced is of the utmost importance as regards launching the mechanisms in charge of estimating the contacted point of the beam. We used a mechanism that was previously proposed in [17] and that is briefly described below.

Consider the coupling torque ${\vec{\Gamma }}^{m}\left(t\right)$ and the force ${\vec{F}}^{m}\left(t\right)$ provided by the F-T sensor in the Cartesian rotated frame ($X^{m}$,$Y^{m}$,$Z^{m}$) of Figure 2b. Denote the effect of gravity on the beam, which is provided in the F-T sensor frame as a force ${\vec{F}}_{g}^{m}\left(t\right)=\left(-m_{b}\cdot g\cdot \sin\left(\theta_{2}\right),0,-m_{b}\cdot g\cdot \cos\left(\theta_{2}\right)\right)$ and a torque ${\vec{\Gamma }}_{g}^{m}\left(t\right)=\left(0,0.5\cdot m_{b}\cdot g\cdot l\cdot \cos\left(\theta_{2}\right),0\right)$, where $m_{b}$ is the mass of the link (see Table 2). Denote as ${\vec{\Gamma }}_{e}^{m}\left(t\right)$ a real-time estimation of the coupling torque in the free movement mode, assuming no gravity. This estimation is yielded by an observer of the first vibration mode, which was experimentally identified as:(15)$$\Gamma_{e,i}^{m}\left(s\right)=\frac{0.37\cdot s^{2}}{s^{2}+0.3\cdot s+16.4^{2}}\cdot \theta_{i}\left(s\right),\;i=1,2$$
where $\Gamma_{e,1}^{m}\left(t\right)$ provides the torque component in the $Z^{m}$ axis, $-\Gamma_{e,2}^{m}\left(t\right)$ provides the component in the $Y^{m}$ axis and the torque component in the $X^{m}$ is zero because there is no torsion in the beam. It is easy to implement this observer, since its inputs are the measured motor angles $\theta_{1}\left(t\right)$ and $\theta_{2}\left(t\right)$. The approximate observation of the coupling torque is completed by adding the torque produced by gravity ${\vec{\Gamma }}_{g}^{m}\left(t\right)$ to ${\vec{\Gamma }}_{e}^{m}\left(t\right)$. Denote the residual error between the measured and the estimated coupling torques as:(16)$$\vec{r^{m}}\left(t\right)={\vec{\Gamma }}^{m}\left(t\right)-\left({\vec{\Gamma }}_{e}^{m}\left(t\right)+{\vec{\Gamma }}_{g}^{m}\left(t\right)\right)$$

A contact is then produced at the instant at which the absolute value of the filtered residual error vector exceeds an experimentally determined threshold. This instant is denoted $t_{c}$.

### 4.3. Estimator Based on Characterizing Vibration Frequencies
(Estimator 1)

The work in [22] demonstrated that the contact point of a sensing antenna could be estimated through the analysis of its natural vibration frequencies after the contact with the object had been established, i.e., by analyzing the transient component $y_{t}\left(x,t\right)$. We should stress that this method was used by assuming that the base end of the antenna is clamped and the point at which the link comes into contact with the object is aligned with the direction of the base end of the link, and that vibration appeared only in the contact direction (in the $Y^{c}$ axis, because the contact point was also assumed to be a pinned end).

Let us consider the dynamic model of Equations (Equation 3) and (Equation 4) described in the previous section, which yields oscillatory responses of the form of Equation (Equation 5). Its natural frequencies can be obtained according to the contact point by solving Equation (Equation 6). Figure 8 shows the natural frequencies of the beam $f_{i}$ obtained by solving Equation (Equation 6) as a function of $x_{c}$. This last equation yields $\beta_{i}$ values, which are transformed into frequency values using the expression:(17)$$f_{i}=2\cdot \pi \cdot \sqrt{\frac{E\cdot I}{\rho }}\cdot \beta_{i}^{2}$$

Figure 8 shows that the relations between the vibration frequencies and the contact points $f_{i}\left(x_{c}\right)$ are not monotonic functions [22]. This means that there is always a range in which two different contact points correspond to a single measured frequency. However, the position of the contact point is defined solely by the whole spectrum of the natural frequencies of the system, or in other words, there is one and only one contact point corresponding to a fixed spectrum of the system frequencies. In fact, knowledge of only the fundamental and the second natural frequencies is sufficient in order to solely determine the contact point if the beam has uniform mass and stiffness distributions.

However, in real applications, $f_{1}$ and $f_{2}$ are estimated from signals provided by the F-T sensor, which include higher natural frequencies and noise. It is often difficult to discover the exact location of the contact point because experimental estimations always include errors. Figure 9 plots the function $f_{2}$-$f_{1}$ parametrized by the contact location $x_{c}\in \left[0,l\right]$. This curve has no self-intersections, signifying that the pair of two first fundamental frequencies make it possible to localize the contact point solely. However, the end point of the sensing curve $x_{c}=0.98$ m is very close to the point $x_{c}=0.615$ m, signifying that the combination of $f_{1}$ and $f_{2}$ is not essentially different and can be confused in practical applications (Figure 9 shows that, in our sensing antenna, a difference of only 1 Hz in $f_{2}$ can produce a drastic change in the estimation of $x_{c}$ between the two previously mentioned values). The information regarding the third natural frequency could be used to make the final decision, but it would highly complicate the procedure. This problem was solved in [22] by adding a small payload to the tip of the beam ($R_{m}\neq 0$). However, overloading the antenna with a tip payload reduces the speed of the antenna and the efficiency of the recognition process.

Frequencies $f_{1}$ and $f_{2}$ can be obtained by recording the measures of the forces and torques provided by the F-T sensor during a fixed time interval that includes several cycles of the fundamental mode. The fast Fourier transform (FFT) of these signals is carried out, and their first two peaks characterize $f_{1}$ and $f_{2}$. In practice, these two frequencies can be obtained by recording a single component of the torque.

Our estimator is different from the method shown in [22] in the following respects:It records the vibration in the direction $Z^{c}$, unlike the method shown in [22], which does so in the direction $Y^{c}$.Our estimator uses only the curve of Figure 8 corresponding to the first frequency, $f_{1}\left(x_{c}\right)$. As stated above, in some cases, it yields two solutions. However, the second estimator will, in that case, discriminate the correct value of $x_{c}$.The motors apply a torque to the antenna in order to carry out the estimation, i.e., in this case, the point at which the link comes into contact with the object is not aligned with the direction of the base end of the link, unlike that which occurs in [22], in which the contact point is aligned with that direction.

### 4.4. Estimator Based on the Torque/Force ratio (Estimator 2)

Estimators of the contact point based on the torque/force ratio were used in [17] and [25]. In this section, we propose an improved version of them.

If Equation (Equation 10) is divided by Equation (Equation 11), we obtain that:(18)$$x_{c}=\frac{|{\vec{\Gamma }}^{m}|}{|{\vec{F}}^{m}|}$$
which makes it possible to estimate the contact point using measurements provided by the F-T sensor. This expression would be true if there were no gravity and the antenna had only a steady-state deflection, i.e., once the transient had vanished.

Since our antenna is moving in 3D space, gravity influences the measured values of torque ${\vec{\Gamma }}^{m}$ and force ${\vec{F}}^{m}$, and its effect has to be removed from these magnitudes. In the presence of gravity, the contact point $x_{c}$ on the beam can, therefore, be approximately estimated by algorithm [17]:(19)$$x_{c}\approx \frac{|{\vec{\Gamma }}^{m}-{\vec{\Gamma }}_{g}^{m}|}{|{\vec{F}}^{m}-{\vec{F}}_{g}^{m}|}$$
where ${\vec{\Gamma }}_{g}^{m}$ and ${\vec{F}}_{g}^{m}$ are provided by the expressions presented in Section 4.2.

As stated previously, the vibrations in contact mode are almost undamped. The time required to reach the steady state and subsequently determine $x_{c}$ from Equation (Equation 19) is, therefore, very long, which makes the estimation procedure unsuitable. We consequently adapted the algorithm of Equation (Equation 19) in order to yield estimations of $x_{c}$ during the transient response of the antenna. During the transient, the measures provided by the F-T sensor are the addition of the steady-state values and the sum of components caused by the vibration modes. In this paper, therefore, we propose to pass signals ${\vec{F}}^{m}\left(t\right)$ and ${\vec{\Gamma }}^{m}\left(t\right)$ through a low-pass filter:(20)$$G\left(s\right)=\frac{1}{1+\frac{s}{\omega_{c}}}$$
whose cutoff frequency $\omega_{c}$ is tuned depending on the range in which the fundamental frequency of the vibrations may vary. This removes the vibration modes from the measured signals and maintains their low frequency components. If we represent the filtered signals of ${\vec{F}}^{m}\left(t\right)$ and ${\vec{\Gamma }}^{m}\left(t\right)$ using ${\vec{F}}_{f}^{m}\left(t\right)$ and ${\vec{\Gamma }}_{f}^{m}\left(t\right)$, respectively, the expression:(21)$$x_{c}\left(t\right)\approx \frac{\left|{\vec{\Gamma }}_{f}^{m}\left(t\right)-{\vec{\Gamma }}_{g}^{m}\left(t\right)\right|}{\left|{\vec{F}}_{f}^{m}\left(t\right)-{\vec{F}}_{g}^{m}\left(t\right)\right|}$$
will yield a reasonable estimation of $x_{c}$ during the transient response of the antenna.

This algorithm is valid only if there is no lateral slip during the contact and ends at the instant $t_{1}+\Delta t_{e2}$ at which the output of Equation (Equation 20) converges to a finite value.

## 5. Results

This section provides our experimental results. Section 5.1 shows the calibration of the $x_{c}$-$f_{1}$ curve, while Section 5.3, Section 5.4 and Section 5.5 illustrate the steps of the estimation method described in Section 4 by showing the processing of the signals obtained in a representative experiment. This experiment moves the antenna on a horizontal plane until it hits an obstacle. The antenna then pushes the obstacle horizontally. Section 5.6 presents the results provided by our estimator.

### 5.1. Calibration of the Fundamental Frequency-Contact Point Function

Estimator 1 determines the fundamental frequency of vibration. This frequency is then used to obtain the contact point from the lowest curve of Figure 8. The curves shown in this figure were obtained from Equation (Equation 6) and critically depend on the parameter $T=\sqrt{\rho /(E\cdot I)}$. According to Equation (Equation 17), this parameter allows $\beta_{i}$ to be converted into $f_{i}$ values. There are two alternatives by which to determine the parameter *T*: (1) from the theoretical values of ρ, *E* and *I* shown in Table 2; or (2) by obtaining an equivalent parameter $T_{e}$, which optimally adjusts the sensing curve $\beta_{1}\left(x_{c}\right)$ obtained theoretically from Equation (Equation 6) to the fundamental frequencies $f_{1}$ obtained experimentally at several contact points.

This work uses the second option for the sake of the precision of the results. An experimental parameter $T_{e}$ is, therefore, determined that minimizes the mean squared error between the fundamental natural frequencies obtained after substituting the $\beta_{1}$ obtained from Equation (Equation 6) in Equation (Equation 17) and the fundamental frequencies obtained from experimentation. In this calibration process, the antenna moved on a horizontal plane and pushed the object with a force whose direction was contained on that plane (the surface of the object contacted was vertical). The fundamental frequency was subsequently determined by carrying out the FFT of the torque measured in the $Y^{c}$ axis. This torque sensed the vibration in the $Z^{c}$ axis, which as previously mentioned, did not have the damping that the vibration in the $Y^{c}$ axis underwent as a consequence of pushing in that direction.

For the calibration of the curve $x_{c}$-$f_{1}$, contact points were considered in a range of $x_{c}=0.3$ m to $x_{c}=0.98$ m, regularly spaced in increments of 5 cm. The fitting of the theoretical curve $\beta_{1}\left(x_{c}\right)$ to the results of these experiments yielded an optimal value $T_{e}=0.2124$. The theoretical value of *T* obtained from the parameters shown in Table 2 was T=0.2282, which implies a relative difference of 7% between these two values. Figure 10 shows the fitting of the curve $f_{1}\left(x_{c}\right)$ to the experimentally obtained frequencies.

The cutoff frequency of the low-pass filter of Equation (Equation 20), $\omega_{c}$, can be tuned using the results from Figure 10. This figure shows that the range of values of the fundamental frequency of our antenna after impact is in the range [31.4,106.8] rad/s and that the lowest frequencies correspond to the contact points that are closest to the base of the antenna. A cutoff frequency of $\omega_{c}=5$ rad/s was, therefore, chosen for G(s), which is about six times lower than the inferior limit of the frequency range. This value allows a reasonable filtering of the vibrations while producing a slight phase lag in the signals measured.

### 5.2. Detection of the Impact Instant $t_{c}$

The impact instant $t_{c}$ is estimated using the residue defined in Equation (Equation 16). The threshold of this estimator is independent of the contact point and the relative orientation between the antenna movements and the object surface. We used a threshold whose value is 0.02 for all the experiments. This was determined by employing the maximum residual obtained during free movements plus a security margin of 50%. Figure 11a shows the measured coupling torques and the coupling torques estimated by employing Equation (Equation 16). Figure 11b depicts the residual error, $\vec{r^{m}}\left(t\right)$, that exists between the measured and the estimated coupling torques, the threshold of the estimator and the estimation of the impact instant $t_{c}$, which is triggered when the residual error surpasses this threshold.

### 5.3. Detection of the Instant $t_{1}$

The time $t_{1}$ is provided by the instant at which the magnitude of the object reaction force $|{\vec{F}}^{m}|$ provided by the F-T sensor surpasses a threshold of $F_{m}^{m}=0.8N$. This threshold value is considered sufficiently high to prevent slipping. Signal $\left|{\vec{F}}^{m}\left(t\right)\right|$ often has a vibration of high amplitude, which makes it difficult to obtain $t_{1}$. This signal is, therefore, passed through the low-pass filter of Equation (Equation 20) tuned with a $\omega_{c}=5$ rad/s, as occurred in Section 5.1, in order to remove that vibration.

Figure 12a shows the evolution of $\left|{\vec{F}}^{m}\left(t\right)\right|$ and its filtered signal. This figure also shows the programmed force $F_{m}^{m}$, which triggers the stopping of the motors when this force is reached. Note that the filter does not completely remove the vibration in this experiment. However, it is important to mention that, in this experiment, the contact is produced at $x_{c}=0.3$ m, which is the worst possible case because the antenna vibrates with the lowest frequency and the highest amplitude. These two features cause the worst behavior of the filter. However, this figure shows that, even in this disadvantageous case, the algorithm yields an estimation of $t_{1}$, which is useful. In any other experiment, the filtering of $\left|{\vec{F}}^{m}\left(t\right)\right|$ is much better, and vibrations do not, in most cases, appear in the filtered signal.

Figure 12b depicts the angle of the azimuthal servo-controlled motor during this maneuver. It shows that this angle is not affected by vibrations and that after the instant $t_{1}$, the motor is stopped.

The coordinate frame attached to the contact point is defined at instant $t_{1}$. The force measured by the F-T sensor at this instant defines the axis $Y^{c}$. The angles of the motors measured at this instant are used to calculate the point $P_{r}$, and hence define the vector $\vec{OP_{r}}$. Axis $Z^{c}$ is subsequently calculated as the unity vector aligned with the cross product of $\vec{OP_{r}}$ and the $Y^{c}$ axis. Finally, axis $X^{c}$ is calculated as the cross product of the other two axes.

### 5.4. Estimation of $x_{c}$ Based on the Determination of the Fundamental
Frequency $f_{1}$

Estimator 1 obtains the fundamental frequency of vibration by calculating the FFT of the moment measured by the F-T sensor in the $Y^{c}$ axis. Figure 13a shows this moment, while Figure 13b shows the magnitude of the FFT applied in an interval $\Delta t_{e1}$ of this recorded signal, in the case of an azimuthal movement and a contact in $x_{c}=0.3$ m. Note that: (a) in this case, axis $Y^{c}$ is aligned with axis $Y^{m}$ of the F-T sensor (in the opposite direction), and the signal provided in this direction is, therefore, $-\Gamma_{y}^{m}$; (b) only one peak is observed in Figure 13b; and (c) this peak corresponds to the fundamental frequency of the vibration of the antenna after the impact, which can be easily estimated from that figure.

### 5.5. Estimation of $x_{c}$ Based on the Determination of the Torque/Force Ratio

Estimator 2 uses the torque/force ratio of Equation (Equation 21) to determine $x_{c}$ from measurements provided by the F-T sensor. As mentioned previously, in order to speed up the estimation process, the measured force and torque are passed through the low-pass filter of Equation (Equation 20) tuned with $\omega_{c}=5$ rad/s. Figure 14a shows the result of 0.44 m yielded by this estimator when the contact is produced in $x_{c}=0.4$ m. Figure 14b shows the result of 0.914 m yielded by this estimator when the contact is produced in $x_{c}=0.85$ m.

### 5.6. Experiments Regarding Contact Point Detection Based on Combined
Static and Dynamic Information

Experiments were performed for different antenna movements and relative orientations between the movement and the object surface. It is not necessary for the object to impact in the direction normal to its surface. However, the angle between the vector that defines the direction of the impact and the vector normal to the surface of the impacted object should not be large in order to reduce lateral rebounds and slipping.

Two sets of experiments were carried out in order to verify the performance of our procedure: (1) the antenna performed azimuthal movements (perpendicular to the direction of gravity); and (2) the antenna performed vertical movements (affected by gravity). In each set of experiments, the antenna came into contact with the object at link points in the range of $x_{c}=0.3$ m to $x_{c}=0.98$ m, regularly spaced in increments of 5 cm. Moreover, five experiments were carried out at each contact point in order to show the repeatability of our proposed estimator. Figure 15a,b show the setups used in the azimuthal and vertical movement experiments.

The procedure described in Section 4 was applied in all the experiments: Estimator 1, which is the most accurate, determined the possible set of $x_{c}$ values (one or two), while Estimator 2 was used to make the decision as to which candidate was the correct $x_{c}$, in the case of there being two candidates.

#### 5.6.1. Case 1: Azimuthal Movement

A first set of 75 experiments was carried out by performing azimuthal movements. Figure 16a shows the estimation of the contact points, $x_{c,e}$, using the proposed algorithm versus the real contact point $x_{c}$. Figure 16b shows the absolute errors between $x_{c}$ and $x_{c,e}$. These figures prove that the proposed technique estimates the contact points of the antenna in a precise manner. These results have a maximum error of 4 cm. However, in most cases, the errors are around 1 cm (1% of the length of the antenna), which shows that the results yielded by our algorithm are significantly better than the results obtained in former works.

Table 3 provides details of the results of this first set of experiments. It shows the real value of the contact point, $x_{c}$, the average value of the estimator for the five experiments, ${\bar{x}}_{c,e}$, the average value of the absolute value of the error of the estimator for each contact point, $\bar{e}$, the maximum error of the estimator, $e_{max}$, and the standard deviation of the error in the estimation, σ.

#### 5.6.2. Case 2: Vertical Movement

A second set of 75 experiments was carried out by performing elevation movements. Figure 17a shows the estimation of the contact points, $x_{c,e}$, using the proposed algorithm, while Figure 17b shows the absolute errors between the contact point $x_{c}$ and the estimated point $x_{c,e}$. These figures again prove that the proposed technique accurately estimates the contact points of the antenna. These results have a maximum error of 6.6 cm. However, in most cases, the errors are around 1 cm, which are again significantly better than the results obtained in former works.

Table 4 provides the details of the results of this second set of experiments. The real value of the contact point, $x_{c}$, the average value of the estimator for the five experiments, ${\bar{x}}_{c,e}$, the average value of the absolute value of the error of the estimator for each contact point, $\bar{e}$, the maximum error of the estimator, $e_{max}$, and the standard deviation of the error in the estimation, σ, are shown.

Finally, Figure 18 plots the fundamental frequencies of the vibration of the antenna obtained in these experiments, and the sensing curve from Figure 10 is superimposed over them.It shows that the $x_{c}$-$f_{1}$ curve calibrated in Section 5.1, which used the dynamic model of Equation (Equation 3) that ignored the effect of gravity, accurately reproduces the $x_{c}$-$f_{1}$ data obtained under the conditions of gravity. This supports the third assumption made about the antenna in Section 3.3, which stated that the influence of gravity on the vibration of our antenna is very small.

## 6. Discussion

The technique that yields the most accurate estimation of the contact point is that based on determining the first two frequencies of vibration in contact mode. However, its application to our antenna had several drawbacks.

The first drawback was that, in many of the experiments, it was not possible to estimate the second vibration frequency. Figure 19 plots the FFT amplitude of the signals $\Gamma_{y}^{c}\left(t\right)$ experimentally obtained for several contact points $x_{c}$. This figure shows that the frequency of the first vibration can easily be obtained in all cases, but that the peak of the second vibration frequency can be observed only when $x_{c}$ is close to the base of the antenna, i.e., when $x_{c}\leq 0.5$ m, which are the most unlikely cases. Moreover, in the case of the second vibration mode, this figure shows that the corresponding peak is not sharp, which leads to an inaccurate estimation of the value of the second vibration frequency. In particular, it is impossible to know where the second vibration frequency is in the FFTs in Cases c and d of the figure.

The second drawback is that a small payload had to be added to the tip of the antenna in order to distinguish between pairs of frequencies $\left(f_{1},\;f_{2}\right)$ that were very close (see Figure 9) and thus avoid errors in the estimation of $x_{c}$. For example, in [22], an antenna of 0.29 m in length was used, and a payload was added to its tip, which increased its total weight by 18.9%. This added payload produced a slight increase in the rotational inertia of that antenna. However, in the case of longer antennae like ours, which is 0.98 m long, that increase in the rotational inertia combined with the torque caused by the gravitational force of this payload in the elevation motor may slow down the movement of the motors. Note that rapid movements are required in order to carry out efficient explorations of the environment. The addition of this payload to long antennae must, therefore, be avoided.

In order to overcome the two previous drawbacks, our method combines the estimation of the fundamental frequency with the estimation of the ratio between the measured torque and force. In order to obtain an accurate estimation of that ratio, it is necessary for the antenna to push the object with a significant force that is greater than the threshold $F_{m}^{m}$. The motors consequently have to provide significant torques to the antenna. This is a third drawback of the method based on estimating the first two frequencies: it cannot be applied to a combined estimation process like ours because it was designed to work under the assumption that the torques provided by the motors are zero in the steady state (see [22]).

A fourth drawback is that [22] obtained the vibration frequencies from measures of the torque in the $Z^{c}$ axis. Oscillations in the $Y^{c}$ direction are, therefore, used to characterize these frequencies. However, in some cases, the pushing force, which is applied in the $Y^{c}$ direction, quickly dampens the vibration in this direction and prevents the attainment of the vibration frequencies from these measurements. This is illustrated in Figure 20, which shows torques in the $Y^{c}$ and $Z^{c}$ directions recorded during the $x_{c}$ estimation process in several contact mode situations. In Case d, the vibration quickly disappears from signal $\Gamma_{z}^{c}$, and in Case b, the first vibration mode is distorted by significantly higher modes that may degrade the estimation of its frequency. However, $\Gamma_{y}^{c}$ provides a relatively clean signal of the first vibration mode in all cases.

Our estimator overcomes the aforementioned drawbacks by combining the dynamic and static information of the F-T sensor:It needs only to estimate the frequency of the first vibration mode.It does not require the attachment of a payload at the antenna tip in order to avoid sensitivity problems caused by the frequency estimation process.The dynamic model of the antenna in contact mode proposed by [22] is extended to the case in which the actuator applies a permanent torque to the base of the antenna.We found that it was not, in some cases, possible to estimate the fundamental frequency from the vibration measured in the direction of the reaction force of the object on the antenna. However, we found that estimating this frequency from the vibration produced in the direction perpendicular to the previous one was more reliable.

Special attention must be paid to the linearity assumption (Assumption 1 about the antenna of Section 3.3) both in free movement and contact mode. Azimuthal and attitude movements in free mode can be approximately linearized and decoupled, as was stated in Assumption 5 about the antenna of Section 3.3. This linearized model yields transfer functions (Equation 15) that are used in the algorithm of Section 4.2 to estimate the contact instant. Moreover, these linearized models were used in the closed-loop control of the free movement in [17,21], yielding satisfactory results. Regarding contact mode, the precision attained in the estimation of the contact point shows that the linearity assumption is adequate. However, it remains as an open question if assuming a nonlinear model of the deflection, like, e.g., in [30], would improve the accuracy of the estimation. This will be the object of our future research.

Finally, we should mention that our methods outperformed the accuracy of the methods cited in the Related Work section. These methods yielded estimates of the contact point with errors of over 3% of the length of the antenna, while our method increases the accuracy of the estimates by more than three times.

## 7. Conclusions

This paper addresses the idea of improving the accuracy of the estimation of the point at which contact is made between an active sensing flexible antenna and an object by combining static and dynamic information regarding the deflection of the antenna with information about the instant of the impact. We show a method that processes these data in order to allow more precise estimations of the contact point than occurred in other previous research (errors of about 1% of the length of the antenna).

Active sensing antennae are being used as an aid for the navigation of mobile robots in dark, dusty or smoggy environments with obstacles. Increasing the precision of the contact point estimation is a step forward, since it endows these robots with the ability to not only detect, but also recognize objects in their surroundings

Our future research will address the reduction in the time required by the estimation procedure and its application to object recognition tasks using a mobile robot.

## Figures and Tables

**Figure 1 sensors-21-01808-f001:**
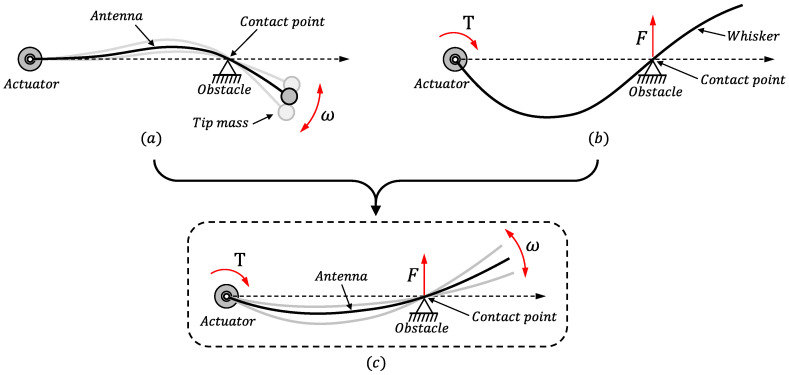
Different methodologies used in: (**a**) [22], (**b**) [23]; and (**c**) the present paper.

**Figure 2 sensors-21-01808-f002:**
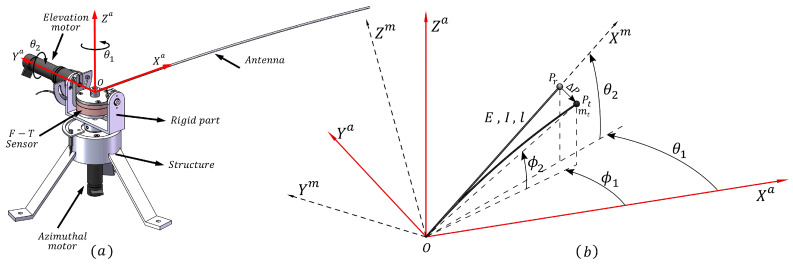
2DOF flexible-beam sensor: (**a**) mechanism design and (**b**) schematic diagram. F-T, torque-force.

**Figure 3 sensors-21-01808-f003:**
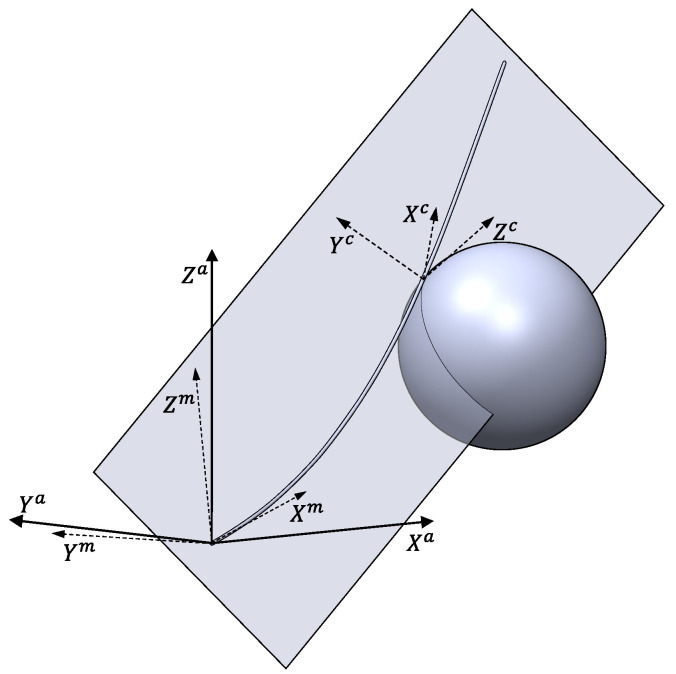
Frame associated with the contact point and the reaction force of the object.

**Figure 4 sensors-21-01808-f004:**
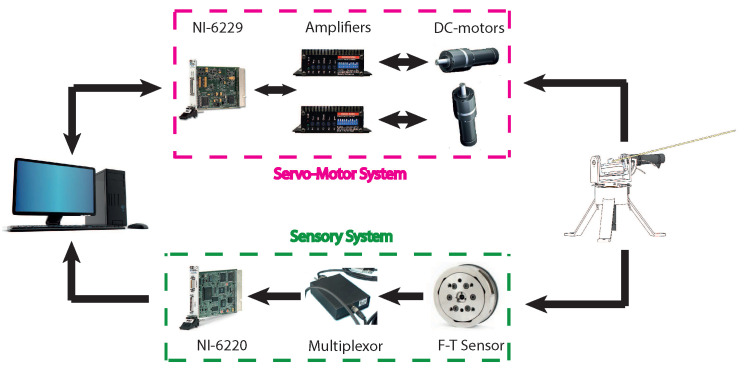
Scheme of the sensing antenna.

**Figure 5 sensors-21-01808-f005:**
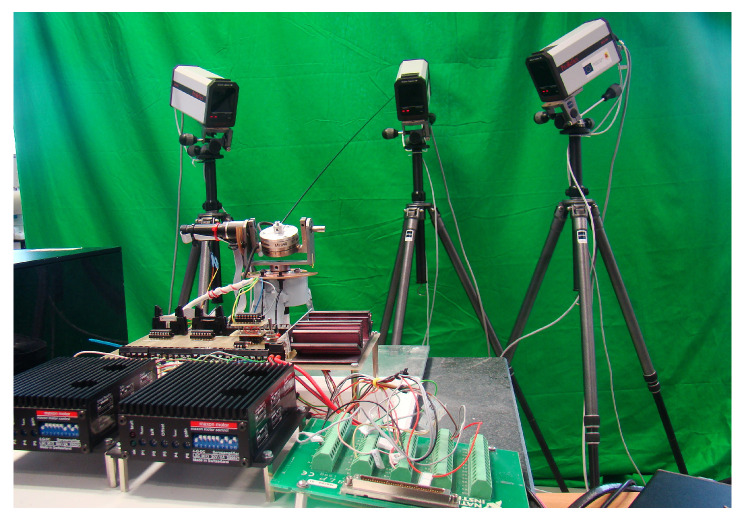
Photograph of the sensing antenna.

**Figure 6 sensors-21-01808-f006:**
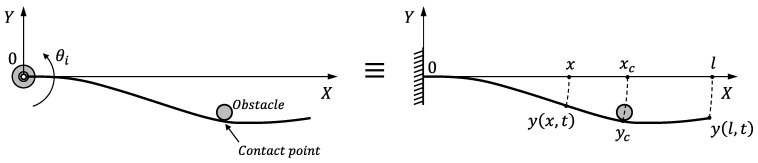
Scheme of the deflection on the bending plane.

**Figure 7 sensors-21-01808-f007:**
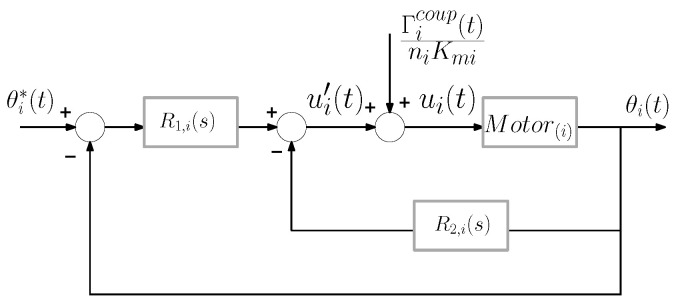
Motor control loop scheme.

**Figure 8 sensors-21-01808-f008:**
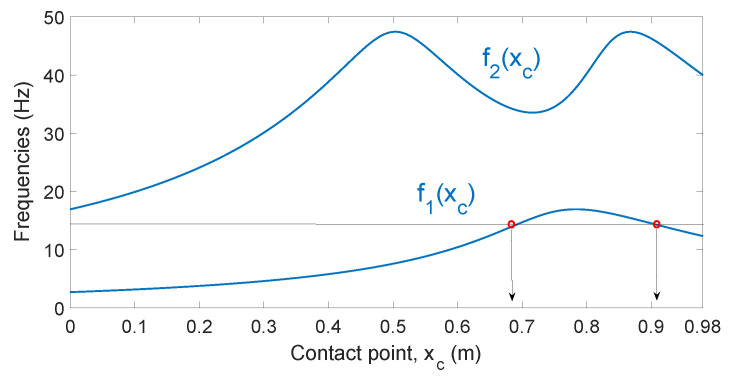
Frequencies of the system.

**Figure 9 sensors-21-01808-f009:**
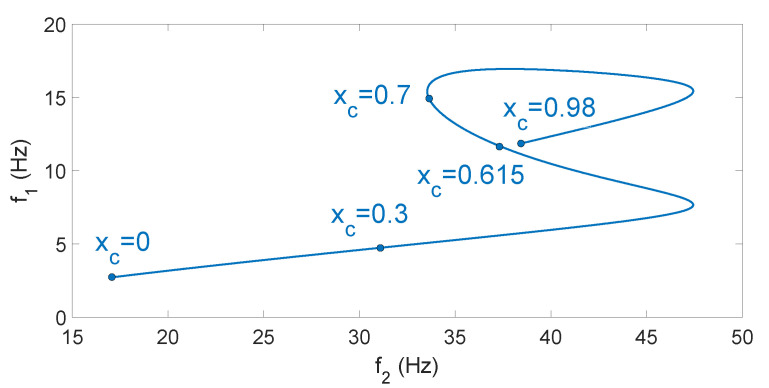
Sensing curve $f_{2}$-$f_{1}$.

**Figure 10 sensors-21-01808-f010:**
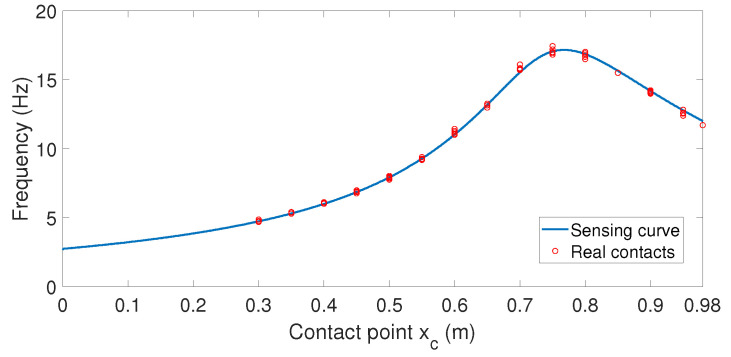
Calibration of the frequency-contact point function.

**Figure 11 sensors-21-01808-f011:**
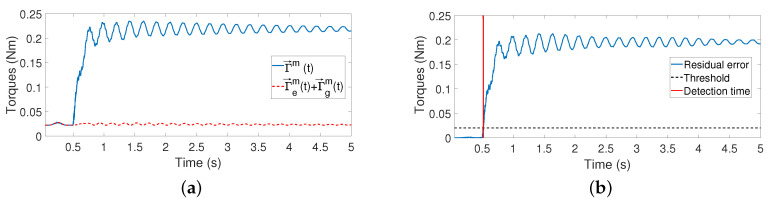
Estimation of the instant of contact: (**a**) Measured and estimated coupling torques; (**b**) residual error.

**Figure 12 sensors-21-01808-f012:**
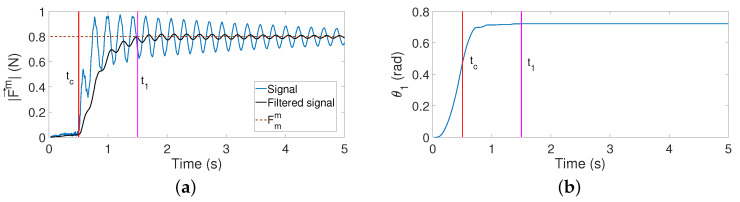
(**a**) Magnitude of the contact force $\left|{\vec{F}}^{m}\left(t\right)\right|$ and its filtered signal; (**b**) motor angle.

**Figure 13 sensors-21-01808-f013:**
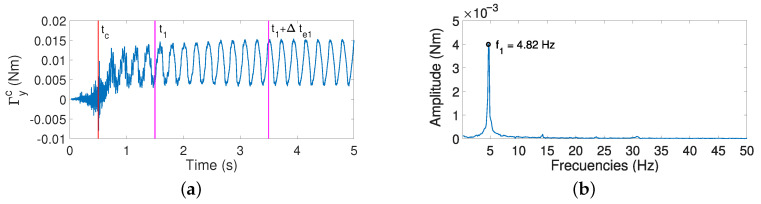
(**a**) Coupling torque $\Gamma_{y}^{c}$; (**b**) magnitude of the fast Fourier transform of $\Gamma_{y}^{c}$ after the impact.

**Figure 14 sensors-21-01808-f014:**
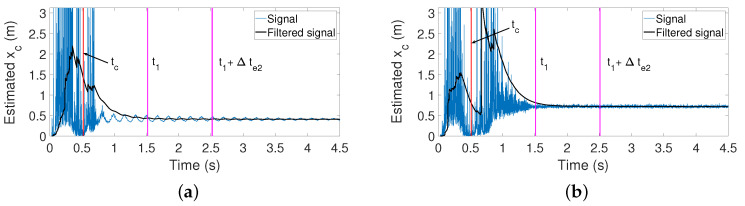
Estimator based on the torque/force ratio: (**a**) contact point 0.4 m; (**b**) contact point 0.85 m.

**Figure 15 sensors-21-01808-f015:**
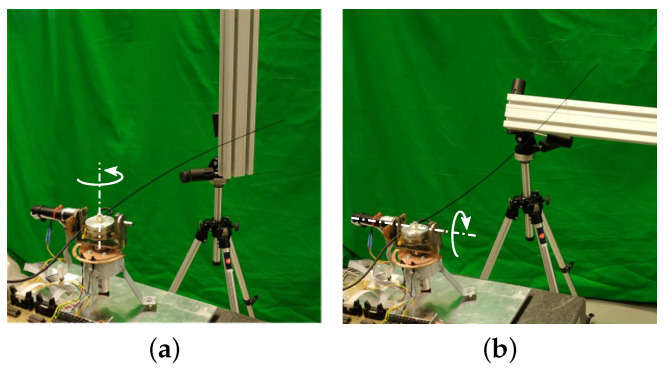
Setups of the experiments of azimuthal (**a**) and vertical (**b**) movements to make the contact with the object.

**Figure 16 sensors-21-01808-f016:**
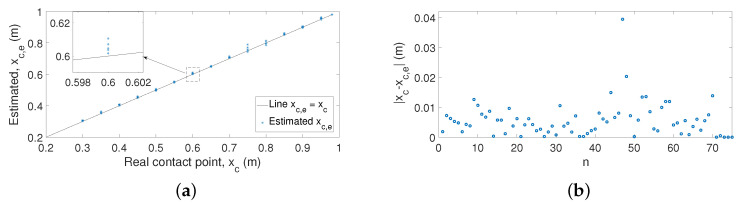
Experimental results obtained for the estimation of the contact point for Case 1: (**a**) estimated $x_{c,e}$ versus real $x_{c}$; (**b**) absolute errors.

**Figure 17 sensors-21-01808-f017:**
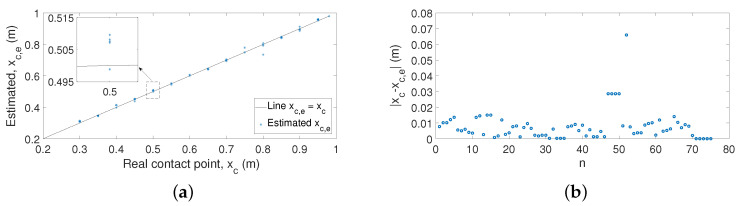
Experimental results obtained for the estimation of the contact point for Case 2: (**a**) estimated $x_{c,e}$ versus real $x_{c}$; (**b**) absolute errors.

**Figure 18 sensors-21-01808-f018:**
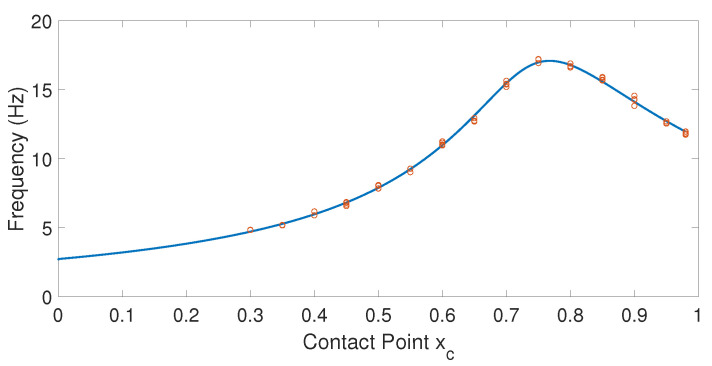
Fundamental frequencies obtained in the experiments of vertical contact versus fundamental frequencies provided by the model (Equation 3) of azimuthal contact.

**Figure 19 sensors-21-01808-f019:**
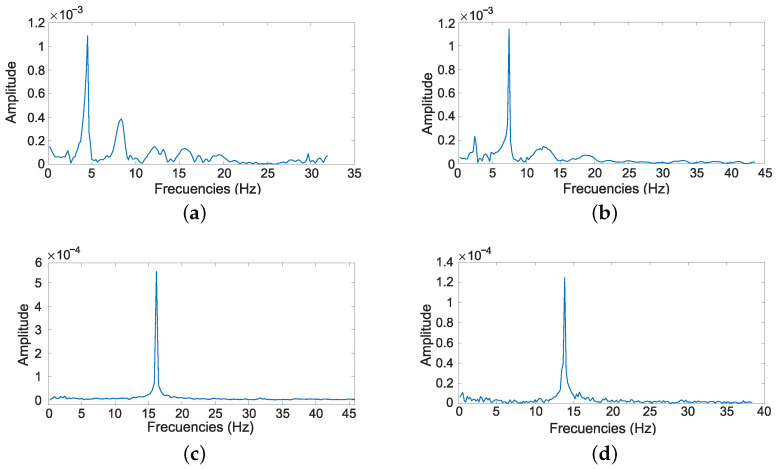
Experimental results obtained after processing $\Gamma_{y}^{c}$: (**a**) contact point 0.3 m, (**b**) contact point 0.5 m, (**c**) contact point 0.8 m and (**d**) contact point 0.9 m.

**Figure 20 sensors-21-01808-f020:**
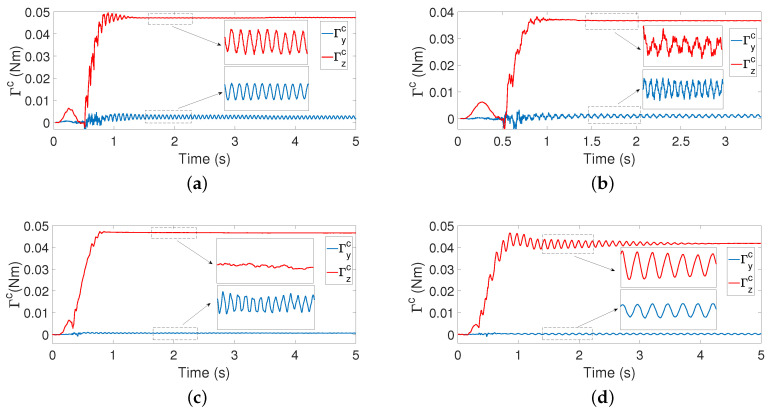
Experimental torques during the estimation process: (**a**) azimuthal movement with contact point in 0.4 m, (**b**) azimuthal movement with contact point in 0.7 m, (**c**) elevation movement with contact point in 0.4 m and (**d**) elevation movement with contact point in 0.7 m.

**Table 1 sensors-21-01808-t001:** Parameters of the motors.

	$K_{\mathrm{mi}}\left(\frac{\mathrm{Nm}}{V}\right)$	$J_{i}\left({\mathrm{kgm}}^{2}\right)$	$\nu_{i}\left(N\cdot m\cdot s\right)$	$V_{i}^{\mathrm{nlc}}$	$n_{i}$
Motor 1	0.003	$6.2\times 10^{-3}$	$30.4\times 10^{-3}$	0.48	100
Motor 2	0.003	$1.8\times 10^{-3}$	$28.5\times 10^{-3}$	0.42	100

**Table 2 sensors-21-01808-t002:** Characteristics of the flexible beam.

Parameters of Flexible Link	Quantity	Unit
Length *l*	0.98	m
Radius *r*	1	mm
Cross-section $C_{S}$	3.08	mm^2^
Section inertia moment *I*	0.785	mm^4^
Young’s modulus *E*	$115\times 10^{9}$	$\frac{N}{m^{2}}$
Linear density ρ	4.7	g/m
Link mass $m_{b}$	4.6	g

**Table 3 sensors-21-01808-t003:** Experimental results obtained for the estimation of the contact point for Case 1.

$x_{c}$ (cm)	30	35	40	45	50	55	60	65	70	75	80	85	90	95	98
${\bar{x}}_{c,e}$ (cm)	30.5	35.7	40.6	45.5	50.1	55.1	60.6	65.0	70.8	76.2	80.0	85.8	90.1	95.6	98.1
$\bar{e}$ (cm)	0.5	0.7	0.6	0.5	0.3	0.2	0.6	0.1	0.8	1.5	0.9	0.8	0.3	0.7	0.1
$e_{max}$ (cm)	0.7	1.3	0.9	1.0	0.6	0.4	1.1	0.3	1.5	3.9	1.4	1.2	0.6	1.4	0.1
σ (cm)	0.2	0.5	0.3	0.3	0.2	0.1	0.3	0.1	0.4	1.5	0.5	0.5	0.2	0.4	0.02

**Table 4 sensors-21-01808-t004:** Experimental results obtained for the estimation of the contact point for Case 2.

$x_{c}$ (cm)	30	35	40	45	50	55	60	65	70	75	80	85	90	95	98
${\bar{x}}_{c,e}$ (cm)	31.1	34.5	41.1	44.9	50.6	55.0	60.1	64.2	69.9	77.3	78.5	84.3	89.6	95.7	98.1
$\bar{e}$ (cm)	1.1	0.5	1.2	0.4	0.7	0.3	0.2	0.8	0.3	2.3	1.8	0.7	0.8	0.7	0.1
$e_{m}$ (cm)	1.4	0.6	1.5	1.2	1.0	0.7	0.6	0.9	0.6	2.9	6.6	1.0	1.4	1.0	0.1
σ (cm)	0.2	0.1	0.5	0.4	0.3	0.2	0.3	0.2	0.2	1.2	2.7	0.4	0.4	0.3	0.02

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
