# Peer review of "Improving the Detection of the Contact Point in Active Sensing Antennae by Processing Combined Static and Dynamic Information"

_sensors, 2021, doi:10.3390/s21051808_

Round 1

Reviewer 1 Report

The authors have proposed a new prototype to estimate the point of contact of the flexible antenna with an object. The work is meaningful and results are instructive. But I still have a few questions before the manuscript is published:

  1. The narration will be clearer if some figures of related works can be provided.
  2. More explanations for Figure. 1(b) should be added to make it clearer. For example, and M do not exist in the figure.
  3. Some abbreviations should be in full when they first appear.
  4. More figures of the proposed prototype added will make it clearer.
  5. Several assumptions are listed in the manuscript. Some explanations should be added how different they are from the reality. Under what conditions can such assumptions be made in reality?
  6. Some parameters should be explained in Eq. (1).
  7. The notation of prime seem to have two different meanings at “” and “”. A new notation to distinguish them will be better.
  8. If the figures of experiments can be provided, it will be more persuasive.

Author Response

The authors have proposed a new prototype to estimate the point of contact of the flexible antenna with an object. The work is meaningful and results are instructive. But I still have a few questions before the manuscript is published:

  1. The narration will be clearer if some figures of related works can be provided.

A new figure (Figure 1) has been added in Section “2. Related work” to clarify the methodology used in the most relevant related works and the relation with the methodology followed in the present paper.

  1. More explanations for Figure. 1(b) should be added to make it clearer. For example, and M do not exist in the figure.

The old Figure 1 now is the Figure 2. Fig. 2b has been redrawn to increase the readability of all the parameters. The representation of the tip mass, M, has been changed to mt, which is zero in our case because our antenna does not have any payload at the tip. The parameter  has also been included in the figure (it was not in the previous figure). It is a vector defined from Pr to Pt which describes de beam deflection at the tip.

  1. Some abbreviations should be in full when they first appear.

The whole document has been reviewed and the words of the abbreviations are now fully written prior to their abbreviated use.

  1. More figures of the proposed prototype added will make it clearer.

In the new Figure 1 included in Section “2. Related work” there is a scheme (c) of the methodology used in this paper to detect the contact point. Also, the new Figure 2a (old Fig. 1a) of Section “3.1. Experimental Setup” has been redrawn, including more explanation of the components used by our prototype. Finally, two photos of the antenna performing the experiments have been included in the new Figure 15, located in Section “5.6. Experiments on Contact Point Detection based on Combined Static and Dynamic Information”.

  1. Several assumptions are listed in the manuscript. Some explanations should be added how different they are from the reality. Under what conditions can such assumptions be made in reality?

The paragraph about these assumptions has been modified and extended in order to include some explanations (in the new version, lines 219-256):

The following assumptions are made about the antenna:

1) The beam deflection is limited to 10% of the total beam length in order to obtain a linear  beam deflection. In this case, deflections y are small compared to the corresponding x values. Then, the deflected abscissa has approximately the same value than the undeflected one,  , and both x and y become the coordinates of a point of the deflected beam expressed in the frame X-Y.                               

This assumption is justified because: a) the figure of 10%, needed to assume a geometrically linear deflection, is an approximate value commonly used in studies of beam deflections and vibrations (a justification of this value can be found in [28]), b) though free movements are performed carrying out fast trajectories, deflections are lower than this limit because the antenna is very lightweight and, then, inertial forces are low and c) in the contact mode, the force exerted by the antenna on the object is programmed to be high enough to allow a reliable estimation of the contact point but low enough to prevent exceeding this deflection limit.                    

2) The antenna has been manufactured to have a uniform cross section.

3) Since the antenna is a very slender beam, rotatory inertia, shear deformation and internal friction are neglected.  

4) The total mass of the antenna is much smaller than that of the contacted object, so that the object is not moved during the sensing process.

5) Regarding the antenna dynamics in the free movement, it was demonstrated in [27] by carrying out extensive simulations and experimentation that, since the linear density of the beam is sufficiently small: a) the vibration associated with the first mode is much more relevant than the vibrations associated with the other modes; b) the Coriolis and centripetal torques are much smaller than the inertial torques and they can, therefore, be neglected; c) the previous item allows us to assume that the azimuthal and attitude dynamics are approximately decoupled, and d) the gravitational torque is significant in the attitude component of the movements.

And the following assumptions are made about the contact mode:

1) The contacted object is rigid.

2) The contact is produced in a single point, as it has been commonly assumed in the previous works.                                  

3) Sliding of the contacting bodies relative to one another is negligible once a specific value of the pushing force has been reached, i.e, if the pushing force of the antenna on the object surpasses a specific value, the slipping is prevented.

4) Since there is no deformation in the X axis of the antenna, longitudinal contact forces do not influence the modelled dynamics.

5) The linear density of the beam  is small enough, and the contact force is big enough, as to assume that gravity neither influences the vibration modes of the beam nor its steady state deflection. In particular, Subsection 5.6.2 checks that the first vibration frequency - which is the one used in this work - does not change because of the effect of gravity.

  1. Some parameters should be explained in Eq. (1).

Sections “3.1. Experimental Setup” and “3.2. Actuator Dynamics” have been rewritten in order to define all the parameters used thereafter in the equations. (in the new version, lines 179-182 and 205-211 have been rewritten)

  1. The notation of prime seem to have two different meanings at “” and “”. A new notation to distinguish them will be better.

The notation of the coordinate frames has been changed in the following way (see lines 165-174 of the new version):

  • Absolute coordinate frame, O-XYZ →  O-XaYaZa
  • Motors coordinate frame, O-X’Y’Z’ →  O-XmYmZm
  • Contact coordinate frame, O-X”Y”Z” →  O-XcYcZc

Now, primes only denote the partial differentiation.

  1. If the figures of experiments can be provided, it will be more persuasive.

As mentioned in Point 4, two photos of the antenna performing the experiments has been included in the new Figure 15 located in Section “5.6. Experiments on Contact Point Detection based on Combined Static and Dynamic Information”.

Reviewer 2 Report

This paper is very interesting. It contains substantial contents on design, modelling, and testing (showing good agreement with modelling). It deserves an acceptance for publication in this journal. However, I have several concerns as detailed below.

1) The literature review should be extended to include more references. The current review is too limited to show a clear motivation.

2) The linear assumption during the modelling is not clarified. It was stated a deflection less than 10% of beam length is used. However, it is not clear if some nonlineararity can be accommodated even though the deflection is still limited to 10% of beam length (see the example paper: planar case: Extended Nonlinear Analytical Models of Compliant Parallelogram Mechanisms: Third-order Models; spatial case: A nonlinear analysis of spatial compliant parallel modules: Multi-beam modules). From the dynamic transfer function, it seems that there is no any nonlinearity taken into account.

3) The design in figure 1 is not clear. A separate part should be shown. How is the beam's boundary condition and geometry as well as desired deformation direction. Is it a free-fixed beam? If so, Fig. 4 does not sound.

4) Most figures have poor quality, which should be redrawn and better labelled. The experimental rig should be explained by proper labelling.

5) English is quite rough, which should be significantly polished.

Author Response

This paper is very interesting. It contains substantial contents on design, modelling, and testing (showing good agreement with modelling). It deserves an acceptance for publication in this journal. However, I have several concerns as detailed below.

Thank you for your positive comment.

  1. The literature review should be extended to include more references. The current review is too limited to show a clear motivation.

In the Introduction, the third paragraph has been included, which deals about applications of robotic whiskers and antennae, in order to justify better the motivation of using these devices (in the new version, lines 40-45):

Lightweight whiskers are being developed with several geometries, and are being used in different applications like object identification and spatial localisation [5]; SLAM for robot navigation [6]; classification of objects based on their material properties [7]; underwater sensing whiskers to measure water flow velocity [8]; whiskers for fluid velocity sensing [9], with the purpose of using them as an aid for drone navigation in dark or turbulent environments, and soft whiskers that actively adjust their morphology in order to regulate their sensitivity [10].

Five new references have, then, been added to the paper: [5], [6], [7], [8] and [10] in the new version.

  1. The linear assumption during the modelling is not clarified. It was stated a deflection less than 10% of beam length is used. However, it is not clear if some nonlineararity can be accommodated even though the deflection is still limited to 10% of beam length (see the example paper: planar case: Extended Nonlinear Analytical Models of Compliant Parallelogram Mechanisms: Third-order Models; spatial case: A nonlinear analysis of spatial compliant parallel modules: Multi-beam modules). From the dynamic transfer function, it seems that there is no any nonlinearity taken into account.

Regarding the linearity in the free movement (in the deflection and in the dynamic response), a fifth assumption has been added in Subsection 3.3 (in the new version, lines 237-243):

  1. Regarding the antenna dynamics in the free movement, it was demonstrated in [27] by carrying out extensive simulations and experimentation that, since the linear density of the beam ρ is sufficiently small: a) the vibration associated with the first mode is much more relevant than the vibrations associated with the other modes; b) the Coriolis and centripetal torques are much smaller than the inertial torques and they can, therefore, be neglected; c) the previous item allows us to assume that the azimuthal and attitude dynamics are approximately decoupled, and d) the gravitational torque is significant in the attitude component of the movements.

This assumption has been used to justify the transfer functions that appear in the algorithm that estimates the contact instant.

Moreover, the linearity issue has been addressed in the Discussion section and a paragraph has been added near the end about this (in the new version, lines 642-651):

Special attention must be paid to the linearity assumption (assumption 1 about the antenna of Subsection 3.3) both in the free movement and the contact mode. Azimuthal and attitude movements in the free mode can be approximately linearized and decoupled, as it was stated in assumption 5 about the antenna of Subsection 3.3. This linearized model yields transfer functions (15) that have been used in the algorithm of Subsection 4.2 to estimate the contact instant. Moreover, these linearized models were used in the closed-loop control of the free movement in [21] and [17] yielding satisfactory results. Regarding the contact mode, the precision attained in the estimation of the contact point shows that the linearity assumption is adequate. However, it remains as an open question if assuming a nonlinear model of the deflection - like, e.g, in [30] - would improve the accuracy of the estimation. This will be object of our future research.

  1. The design in figure 1 is not clear. A separate part should be shown. How is the beam's boundary condition and geometry as well as desired deformation direction. Is it a free-fixed beam? If so, Fig. 4 does not sound.

Old Figures 1 and 4 have been redrawn and relabelled as Figure 2 and Figure 6. In Fig 2a, a proper labelling of the robot parts has been carried out. Fig 2b has been redrawn and the explanations added to the text have clarified its meaning. Fig. 6 has been modified in order to make clearer its meaning, and now includes two representations: the first one schematises the robot in a contact manoeuvre, while the second one represents the equivalent situation in the contact mode, in which the antenna behaves has a free-fixed beam.

  1. Most figures have poor quality, which should be redrawn and better labelled. The experimental rig should be explained by proper labelling.

As it has been indicated above, new Figure 2a has been redrawn: the different parts of the sketched robots have been properly labelled. All the graphics included in the paper have been redrawn to increase their readability, e.g, the size of the labels has been augmented in many figures.

  1. English is quite rough, which should be significantly polished.

English has been revised by a native English speaker.

Reviewer 3 Report

Dear authors,

I have read with interest your paper. It is very well written and the presented results could be very helpful in practical applications. Moreover, looking at the references, it seems that the authors are experts in their fields.

I just have some comments to clarify/improve some aspects:

  1. line 149: you wrote "the equivalent length of the beam is l", but "l" is not specified in Fig. 1;
  2. In Fig. 1, coordinates Φ1 and Φ2 do not seem to be defined w.r.t. the absolute frame O-XYZ. Please, consider to improve the readability of Fig. 1b;
  3. Table 1: what is ni? It is not specified in the text before the table;
  4. line 165: you talked about "mechanical limitations", without giving any specifications. Could you please explain what there mechanical limitations are?
  5. Eq. (1): what is θi?It should be specified before the equation;
  6. lines 220-222: you wrote "Here, and throughout the paper, dots are used to denote time differentiation with respect to time t, and primes denote partial differentiation with respect to coordinate x", but this sentence should appear before Eq. (1);
  7. Eq. (4): it is not specified before this equation what "xC+" and "xC-" are;
  8. line 437: you wrote "increments of 5mm", but at line 504 you wrote "increments of 5 cm", so what is the correct unit of measurement?
  9. the caption of Fig. 16 cannot be simply " Experimental results", you must specify of what;
  10. line 543: you wrote "FFT of the signals Γy(t) experimentally obtained", but more correctly you show just the amplitude;
  11. even if I did not detect auto-plagiarism, please avoid to reuse exactly the same figures from other papers of the same (or some of the) authors. Fig. 6 is exactly the same as Fig. 4 in Ref. [16]. Maybe you can redraw it or refer to Fig. 4 in Ref. [16].

Author Response

I have read with interest your paper. It is very well written and the presented results could be very helpful in practical applications. Moreover, looking at the references, it seems that the authors are experts in their fields.

Thank you for your positive comment.

I just have some comments to clarify/improve some aspects:

  1. line 149: you wrote "the equivalent length of the beam is l", but "l" is not specified in Fig. 1;

The old Figure 1, relabeled as Figure 2, has been modified and now includes an “” in a clearer manner.

  1. In Fig. 1, coordinates Φ1 and Φ2 do not seem to be defined w.r.t. the absolute frame O-XYZ. Please, consider to improve the readability of Fig. 1b;

The old Fig. 1b (now Fig.2b) has been redone to increase readability of all the parameters.

  1. Table 1: what is ni? It is not specified in the text before the table;

The definition of the reduction gear ratio  has been included in the paragraph above Table 1 (in the new version, lines 179-182).

  1. line 165: you talked about "mechanical limitations", without giving any specifications. Could you please explain what there mechanical limitations are?

It was a typo: the comma of that sentence must by a colon. In the new version, the sentence is “….owing to mechanical limitations: the torques of the motors, the….”.(see line 185 of the new version).

  1. (1): what is θi?It should be specified before the equation;

θi is the angle of the motor . An explanation of the new parameters that appear in Eq. (1) has been included in the paragraph below the equation. (see lines 205-207 of the new version).

  1. lines 220-222: you wrote "Here, and throughout the paper, dots are used to denote time differentiation with respect to time t, and primes denote partial differentiation with respect to coordinate x", but this sentence should appear before Eq. (1);

That sentence has been placed in the paragraph previous to subsection “3.2. Actuator dynamics” label, i.e, previous to the presentation of all the dynamic equations. (see lines 198-200 of the new version).

  1. (4): it is not specified before this equation what "xC+" and "xC-" are;

The explanation of the parameters and  has been included after Eq. (4). (see line 271 of the new version)

  1. line 437: you wrote "increments of 5mm", but at line 504 you wrote "increments of 5 cm", so what is the correct unit of measurement?

The correct unit is 5 cm. It has been changed and now appears correctly. (see line 489 of the new version)  

  1. the caption of Fig. 16 cannot be simply " Experimental results", you must specify of what;

The caption of this figure (now Figure 18) has been changed to: “Figure 18. Fundamental frequencies obtained in experiments of vertical contact versus fundamental frequencies provided by model (3) of azimuthal contact.”.

  1. line 543: you wrote "FFT of the signals Γy(t) experimentally obtained", but more correctly you show just the amplitude;

It has been modified and now that sentence is: “FFT amplitude of the signals Γcy(t) experimentally obtained".(see line 597 of the new version)

  1. even if I did not detect auto-plagiarism, please avoid to reuse exactly the same figures from other papers of the same (or some of the) authors. Fig. 6 is exactly the same as Fig. 4 in Ref. [16]. Maybe you can redraw it or refer to Fig. 4 in Ref. [16].

This figure (now Fig. 7) has been redrawn, though the meaning is the same as before.

Round 2

Reviewer 2 Report

The paper has been addressed well